# Performing Venice's Stones: *Vedute Manoeuvre Redux*

## Heather H. Yeung

School of Humanities, University of Dundee, Dundee DD1 4HN, UK; h.yeung@dundee.ac.uk

**Abstract:** 'Venice excels in blackness and whiteness; water makes commerce between them'. So writes Adrian Stokes, in his 1947 study of the city, its architectures, and its art. This very sentence performs a problem of Venice that has vexed those who have made art, literature, and other writing of the city, in the city, from the city: Venice asks us to take its measure, its shadows and light, its water and stones—but this is even more complex than a chiaroscuro, 'commerce', aesthetic and economic, plays with what is clear and what is not, tipping us between registers we fail to fully comprehend. And thus we are brought too often to perform and replicate such confusion and inability to 'account for' the polytropic, polymaterial, and polytemporal registers the city simultaneously operates upon, or 'makes commerce between'. And yet there is an artistic method that can account for the strange and often highly problematic spoliate economies of Venice, a method which also bridges walking practice as political performance art, and situated performance as art historical practice. This is a poetic-performance method that is provided by the artist Tim Brennan's Vedute Manoeuvre, first performed in the Venice Biennale 2011, and re-performed as part of the research work documented here. Vedute Manoeuvre, I claim, is a method whose polyvocalic polyvisual modes, whose art-act as common experience and experience of the complexity of the artistic and architectural commons and commerce of Venice, is perhaps the only way of 'giving voice to' the polytropic, polymaterial, and polytemporal problems we encounter when we encounter Venice, its water, and its stones. We thus re-orientate the multiple other ways that spoliate, colonial, archipelagic Venice has been found difficult in previous attempts to perform an accounting of (and, indeed, of artistic commerce with) this vexed and vexing city, with Vedute Manoeuvre as invitation toward a performance 'redux', as crux and as solution. The work presented here—an essay in the truest sense—is also a mode of performance which demonstrates in its own attitudes to the question of the manoeuvre the act and art of manoeuvre itself.

**Keywords:** Venice; performance; manoeuvre; artists book; spolia; redux

## 1. On 'Redux'

Venice, spoliate city. Here –

The dash gestures. Need I write more than it implies, for already, I wager, with the first word of the first sentence your mind, my reader, has veered along innumerable remembered visual, artistic, sonorous, musical, literary, lived and imagined canalways and lagoonpaths, across stone campo and stony passageway, as the city itself contains. Already, with the ghosts, perhaps, of biennales attended, ideas of the carnivalesque, Ruskinian remembrances, you have reconstructed for yourself that it may mean to perform, or to perform again, Venice's stones, its spoliate formulations. 'Here—'. The dash acts as an imaginary apostrophe, an invocative revocation, a typological representation of the force of the Warburgian *pathosformel*, or a Benjaminian *gestus*.[1] Looking back to 'Venice' cast here in letters, but, in the gesture of the antecedent dash, (t)here, back and forth, in imagination and time, we appear at once supremely mobile within a specific sort of 'time of art',[2] and also (or even, thusly) engaging in an active mode of critical hesitation between a surfeit of possibilities which must, within a certain logic of performative multiplicity, demand a navigation of such simultaneities whose complicated intersections an interaction

consistently debunk the possibility of a simple 'time of art'; what Nagel and Wood have called a mode, not of backward-only-looking or fetishisable 'anachrony', or 'anachronism', but of observation of the 'anachronic'.[3] 'Here—', recapitulation exists alongside temporal reduction, and expansion, irruption or careful interruptive counterpoint; critique.

　　　　　　　　　　　　　　　　　　　　　　　　　　　　　　　　– or, here—

The point of aesthetic and conceptual oversaturation which, with a given tipping point may become fetish, but ought rather to give rise to critique, precedes this essay by centuries, and so, in some sense, loses a singular origin point, or the punctum of a singular 'artist-function' upon which any of this ventriloquial Venice-work might be grounded. So we dwell in the gesture of the hyphen, the dash, the mark which at once joins together the disparate, demonstrates otherwise non-visible betweens, and propels us in two opposing directions in time, which concretises and, a typographer's or coder's nilling, or a mathematician's minus, negates. How bored are we of Venice? How often do we assume its operations as cultural form? Fluvial, the dash (or Venice) 'dashes' away; 'dashes' the water against the rocks, the light against what is solid, reflects, recapitulates. Attempts to dispel a hyperromanticised Venice, whose accreted myths and muthos are held in the interaction of stony foundations with lazy lagoon water, performs on us its own performance;[4] in which dispelling it meets, consistently, a dreamwork or dreamtime, a zombie-city existing in a material 'half of life',[5] whose intensities of architectural mesalliances force their synecdochic transfusion into human practise. We 'read' Venice, and find ourselves in a spoliate mode of re-production, a cobbling-together of material and immaterial incongruities, re-cognition of the composition that is navigated when the idea and reality of 'Venice' is (simultaneously) navigated, and when navigation, thus, becomes a necessary condensation and performance (we will later see, through a consideration of the *Vedute Manoeuvre*, how such a performance exerts a multiform critique and exposition of this spoliate navigation, which is a part of the idea of the 'redux').

　　　　　　　　　　　　　　　　　　　　　　　　　　　　　　　– and, here—

Indeed, the very foundational myth of the city (of course, much repeated, apocryphal!), based on a vision of St Mark in the pre-city marshy lagoonland, is such a corrupt or spoliate dreamwork which nevertheless is brought forward into the names of places and buildings, and given as divine sanction reason for certain forms of exploitative trade. That great modern English commentator on Venice, Adrian Stokes, was right to foreground three primary access points to Venice (which of course must needs occur simultaneously and in a sort of anachronic, thus inherently dialectical, relation with each other): matter (which is to say, occlusion, chiaroscuro), water (which is to say communication), and trade (which is to say economies, which is also to say movement and material change): 'Venice excels in blackness and whiteness, the water makes commerce between'.[6] Excelling not only in trading (itself), thus in certain economies, condensations, and strange deprivations, but also in the anachronic, Venice, calls, too, outside of itself for its self-instantiation in the mind's eye which frames and internally pauses in the aestheticisation process of its own viewing practise: the very foundations of the city (of course, eminently and irrevocably material), composed of the strikingly pale and internally shadowed *kirmenjak* or *pietra Istria* quarried on the Dalmatian coast which Ruskin so comically likened to a Swiss cheese, are imported,[7] essential to Venice, and absolutely non-Venetian; the city's skyline and central architectural alignments composed, romanticised, and re-composed as a series of acts of *imitatio Constantinopelos*.[8] How, then, do we see what we see? The city (per)forms itself through the public display, artificialised not only by so many years of carrying the footfall of carnivals, of the carnivalesque—Bakhtinian—of biennales, by the irreligious re-formations of so much visual chatter or tourist-pull. We are wrong to think that these aspects do not work together, but must make of them not glut, nor pilgrimage, nor postmodern case-study, but rather a work, a lesson, and thus a practice. A-temporal, polyvalent, and yet place-based and unquestionably durational. So we return, origin-points dashed, even the stony limit questioned due to its spoliate display. To lead (ducere) back, again, or about (re-); to bring back enhanced rather than reduced (but to reduce aspects to a portable essence);

a triple-fold movement; a multifold Proustian 'trip';[9] this, then, is the idea of the 'redux'.

or, again . . . –

## 2. Exposition/Situation

It is August 2022, and I have returned to Venice by the old routes, from the Dalmatian Coast of Croatia. Like the *pietra istria* itself. As in Thomas Mann's *Death In Venice*. Literary/materialist indeed. The city is approached by ferry. The Adriatic, even in summer, is turbulent. There are a few of us—a poet, a philosopher, a critic, and an historian.[10] We sound like the beginning of a joke; provide illustration of different modes of Venetian pathos formulae, or, how the city's variant singularities lend themselves to the kaleidoscopic (an academic may posit, 'interdisciplinarity'; the kaleidoscope lends itself better to the shifting, colliding, movements of mediated vision). In transit, each of us has various meditative plans of a 'romantic' 'approach' to Venice scuppered by the consistent reminder that the ferry is forcing its way too quickly through the waves against the water's will, which is a mode of transportation to which the body is not yet adapted. There is little conversation, and many closed eyes. The often-romanticised voyage is, rather, a blank. Reaching the bounds of the city, the boat moves through designated lagoonways where the city's energy, waste, and trade infrastructure juts into the water on obvious display. The blank populates itself, first, then, with the obvious work of a distinctly twenty-first century 'commerce'. There is a container ship 'AS Beauty'. Cormorants give way to seagulls on the pontoon-posts which cast shadows rather than reflections on the heavy water. Buildings. Smaller boats obviously at work rather than leisure. A decreasing number of natural shorelines. A decreasing number of trees. Leisure boats. Flashes of white stone against brickwork. The much-coveted, much-reproduced *imitatio Constantinopelos*, seen for 10 min at odd angles to the boat's approach and small enough through tricks of distance and Lilliputian perspective to seem, to the finger held up to the skyline, to be easily plucked up and removable, Lego-like, and as soon seen as then disappeared, or (momentarily at least) dashed away.

Venice, and the dash which is its precedent and antecedent, perform the act or art of the 'redux', which is a methodological key to this essay, and to, I argue, a way of looking *with* (at, through) the profoundly interdisciplinary material-immaterial performance practices of the artist, poet, and thinker, Tim Brennan. In turn, the art or act of the 'redux', the dash, and Venice, provide a way of appreciating the polyvalence of Brennan's artistic action in the method of the 'manoeuvre' that demonstrates effectively the ways in which the practise of the manoeuvre generates not only aesthetic practice but also cultural, historical, and political critique. The plan to revisit Venice and the *Vedute Manoeuvre*, which is a work designed by Tim Brennan for the 2011 *Venice Biennale* collection 'The Knowledge' curated by James Putnam, is one that had been long planned by this writer, and long put off by a series of other engagements and the effects upon travel of local and global affairs. And the art of 'redux', here, perhaps predictably also has deeper roots. Revisiting *Vedute Manoeuvre*, whether distantly or in situ, is also to revisit one of my own early works, in which I seek, alongside Brennan's strong situational and durational performance practise, to articulate the possibilities of the essay form as (ad)venture into a similar, or simultaneous, performed space. A mode of 'live' commentary, if you will, which took the name 'Venice Ventriloquised'. The text of that work forms a part of the performance artefact of *Vedute Manoeuvre*;[11] 14 8 × 6" double-sided cards, which on one side contain a print of a Canaletto painting or sketch with hard white surround, and on the other, a series of texts (black and dark grey on white): the numerical indicator of a viewing 'station', a performance or manoeuvre indication (highly place- and orientation-based), a quotation from a Venice-inspired work, and an extended phrase from 'Venice Ventriloquised'. In returning to Venice in the act of *Vedute Manoeuvre redux*, therefore, through even the *disjecta membra* collated by Brennan in the work's material (rather than performance) form, I would be encountering the anachronic practises not only of Brennan and Venice, but also of myself.

There are already many ironies here, as well as possibilities for partisanship of gesture in performance practice, for things to fall through the gaps which mark the start and end of any part of a spoliate formulation. Indeed, resonance is added to the redux if one considers that in its first (*Biennale*) performance, the *Vedute Manoeuvre* was neither performed in full nor in the sense proscribed by the work itself, rather, the timing of the Biennale and the demands of the exhibition sponsors were such that the work began with its starting-point obscured (since the performance-material of the work begins with the image of a *camera obscura* this is perhaps not as denaturing to the work as it may seem, rather, a deliberately occlusive framing device), and subsequently performance-participants, cards in hand, were taken by barge to some locations, in no particular order; the artist performed parts of the work in the *medias res* of the work itself.[12] When the *Vedute Manoeuvre* was next displayed, as part of an exhibition on Art Walking at the Northern Gallery of contemporary art, it was not performed at all—the cards were displayed in order on a shelf, gallery policies prohibited their being handled and read (or performed), there were no co-respondent *vedute* for the cards' images and texts to 'speak' to and within, rather, the white wall of the exhibition space, as a viewer promenaded from the start to the end of the display. The time of art in performance becomes one whose reliance on hermetic visuocentricity (here, one is only looking at reproductions of a series of Canaletto images from the Royal Collection, which one may as well visit in the Royal Collection) expropriates the work (and thence time) of the artwork.[13] A large part of the labour as well as the dialogic force which is a principle aspect of the work disappears in its interaction with the conventional economies of curation and proprieties of display. In such contexts, the material form of the work engages in its own dissolution, whilst simultaneously giving us the necessary information such that we can engage with the entire situation necessary for its full practice.

By 'situation' here, I mean the sort of temporal and durational extension that Brennan, in his development of the *manoeuvre* as performance method, demands of his interlocutor-participants. The 'situation' of the work is, of course, polyvalent, multiform (I have elsewhere written—drawing on the work of Julia Kristeva and of James Joyce—of its kaleidoscopic or verbivocovisual nature).[14] The practice of the work is visual, textual, vocal, performative, material, place-based, durational, political, and self-historicising, always allowing for intersections between each of these aspects, for the *gestus*, or dash, to draw the voice or eyes towards a different part of the commons the *manoeuvre* navigates and creates, towards different multivalent possibilities of choice and critique. Through the life of the work in performance, we lose the *imitatio* and gain the active art of *reducere*. The concept-practice 'manoeuvre' itself calls to this essential dynamism of condensation. In *Vedute Manoeuvre* such practice exhibits itself a series of valences, as follows.

## 3. Base Materiality

First we must disambiguate the material 'work' of the *Vedute Manoeuvre*, which is card-held, and exists at once as the performance commands, an artefact which must be mobilised and incorporated within the performance, and the performance documentation, from the *manoeuvre*, which is the method and performance itself. *Vedute Manoeuvre* is comprised of 14 cards. The cards are double-sided, containing an image and text. The image is itself double-sided before it is de-doubled through its situational emplacement and 'work' within the *manoeuvre* itself: it is with the exception of the 'camera obscura' card (Station 1)[15] an art-quality reproduction of a Canaletto image of Venice with a thin white border. In this way the framing (image/border) mirrors the visual shock-effect of the stark white borders of the plates in Stokes's mid-period *Venice*,[16] which in turn mirror the way Venice's stark white *pietra Istria* foundations frame the shadow and brick of the city. Already in these cards a way of seeing has been materially condensed—the work of stone which is so little 'seen' except peripherally is made a peripheral to visual representation—in which condensation the practise of the *manoeuvre's* visuocentric critique begins.

Such critique continues through the simple effect of the use of the oversized, double-sided, card as the work (and part of the work). In situ (of which more later), our decision

regarding which 'side' of the card to 'read' as part of the performance of the *manoeuvre* is continually disrupted by the un-handiness of the cards and the card-pack, and the gloss on the print further mirrors the visually disruptive commercial 'flash' of water as the cards buckle slightly when held up within the performance of the *veduta*. Thus, an essential element of the performance of the *manoeuvre* exposes itself: that it interacts critically with the sort of exemplum-based decision-making which gives 'point of view' (or *veduta*), and which is broadly conceptually shared by a series of performance modes that Brennan's work critiques and which in thinking of the performance-aesthetics of Venice are condensed into a sort of historical materialism: the historical, the guide-book, the aesthetic, the exegetic, the ecclesiastical, the architectural, the ekphrastic, the literary . . . We must choose what we see when we look at the cards; we must choose to have our perspective governed by the Janus-faced visual or textual, but are also reliant also on the relation between these terms (what is held in the card) and the haptic (how the card is held, and its material form 'managed' in performance). The choice is akin to that of a post-card in its intermedial double-sidedness, should we wish to assess the *Vedute Manoeuvre* cards in such a manner, whose address is multiply gestural, open, and occlusive; thus, one aspect of their performance method.

## 4. Locative Matters

The *Vedute Manoeuvre* cards demand that one places oneself in a series of locations in central Venice (the 'stations' of the *manoeuvre*) and spends time moving between these stations (the action of the *manoeuvre*) in the order demonstrated by the cards themselves (station numbers are given on the 'text' side, but consolidated by the image 'mapping' onto the perspective the station offers onto Venice), that is, if once chooses to follow in one's work of the performance (the re-dux) the order the cards suggest.[17] The materiality of the work is thus multifold, and its Venetian location as foreign to itself as to Venice are its foundational *pietra Istria* or constitutive spolia. The practice of *reducere* when standing at one of the relevant 'stations' is one of choice-predicated condensation—we choose how far to turn our card-based kaleidoscope—which is the work of the work, whose display occurs in part through our manipulation of the card work in situ, and is vastly deranged and further concentrated through the way that the performance of the work relies on the manoeuvre through public spaces and intricate positioning to 'find' the correct alignment of material conditions suggested by each card in our self-stationing.

Each 'station' suggests we take such a station—station ourselves; stop—at a given (public) spot (as a guide-book may suggest an optimal view-point, or as Canaletto manipulates the art of the *veduta*), which—for we are, perhaps too predictably in Venice in the height of the summer, in search of the high sun's short shadows—adds an odd aspect to the work's performance: it immediately involves participation in an explicitly touristic infrastructure. The work is at once in the time of the touristic passage (the time, in the c.21st, of the selfie-and-move-on) and a rock in the flow of this instagramming-river, as it demands a certain time of performance at each station, and the work of the work is independent of these economies, whilst also drawing (through difficulties in navigation) attention to this flow. The way that the 'station' also calls to the time of the meditative-ikonographic practice of station as stopping point in a Catholic via frames and flattens the images in yet another temporal mode. People peer over one's shoulder as one holds a card up to 'match' Veduta against the view from the station; the material nature of the cards calls into question forms of authenticity, forms of worship: the perspective of *veduta* (real) on *veduta* (card-held) forces us to notice in the brain's strange game of dis-orientation-and-spot-the-difference which is not only historical but also aesthetic. The performer of the *manoeuvre*, held in the public flow, does not follow the flow-pattern; begins through the act of the *manoeuvre* to estrange themselves from their surroundings by using an odd screen to their vision, to notice with further attention thus the oddness of their own movements and other acts—beforehand with time we already possess our images, our *souvenir*, in the material presence of the 14 cards, whose insistent commanding presence in the act of the *manoeuvre* disrupt the unidirectional flow of time from experience to object-memory, rendering already

anachronic our subject-position on the debatable grounds of what is commonly (Venice) and uncommonly, yet nevertheless in the common space of the performance, (the *Vedute Manoeuvre* cards) held. The manoeuvre, as it were, opens up through, within, in spite of, what is held in the hand, and that which is held in the hand is the matter which is at once the predicate, the demand, and the distraction from the *manoeuvre*'s *other* matter at hand, or, handiwork: Venice.

## 5. In-Voices-Between

Each of the 14 stations represented on the articulate cards is different, and is related to different spaces in central Venice. Yet the *manoeuvre* begins before the first station. The walkers walk together in the time of the manoeuvre in a mode of 'towardsness' (moving *to* the designated station; performing the command of emplacement each card asks for). Yet these mo(ve)ments towards and away from each station are noisy—full of the touristic commons of Venice. Brennan's *manoeuvres* allow their performers, in holding simultaneously the shared practice of the *manoeuvre* and the excesses of the material, auditory, and visual space in which the *manoeuvre* takes place, to move in and out of a mode of self-reflexive performance or meditation and a quotidian in which a forgetting-of-art in conversation may also occur. In tourist-ridden central Venice, the spaces between stations are noisy with sonorous as well as visual distraction in a piecemeal of languages and exclamations. What then are the noisy spaces in the *manoeuvre's* movements? What are the gaps 'between' in which a different focus is built, where the process of dynamic distraction and association takes hold in which the aesthetic-purposive gives way? What does it mean to be simultaneously placed 'between' points and also a central node in an action?

Writing on Brennan's *Luddite Manoeuvre* (2008), the walking artist Misha Myers observes these apparently interstitial moments as constitutive of an 'art of conversive wayfinding':[18] walkers (as for Myers the manoeuvre-performers are engaging in a *walking art*) create hubbub, are attuned to noticing the hubbub around them due to the engagement with the *manoeuvre*, about which they may talk, gesture away from, be distracted by. Myers reads this as a mapmaking process, conversational pathmaking, a commons of the walk-plus-talk, a making in-common(s) of directional agency 'between' various waypoints. The *manoeuvre*, of course, adds to this—given certain contingencies—as it allows for critique to be performed para- to, punctuated by, and in development with the material-stationary and visuo-textual aspects of the work; the noisy interstices are made visible, foregrounded. As Janet Hand writes, this is not, however, a state of homogenising group-think bestowed upon participants by the fact of their engaging with the *manoeuvre*: the 'disparate elements of the walks rely on each walker's capacity to make, edit, unmake, and connect . . . '[19] which capacities ebb and flow of course in no particular order. The work of the *manoeuvre* is one in which the question of the ownership of the work, its *res publica*, is continually raised in this mode in which the work simultaneously un- and re-makes itself through un-performance (performing the movement between staged or choreographed performance moments). The before of the work's voicings shapes the perspectives which the 'station-based' aspects of the work condense and reframe in the fleeting commons of the performance-choice of a single quotation-reading on a governed spot.

These are and are not just conversations, fleeting, throwaway, and contingent, but also through the framing by the act of *manoeuvre* within public spaces call attention to the voice and how it works, gestures, points, consolidated by the text-performance to come and just departed from. The interaction of levels of material and voice shows how the voice and vision interrupt each other continually in framing, how both interrupt how we listen and conceive of a 'hearing' practice, how both add to the *manoeuvre* the question of vocality and may provoke moments cognate with what Pauline Oliveros calls 'deep listening'.[20] The matter of the (object) voice is one which, as Mladen Dolar remarks, possesses a topology which 'dislocates it in relation to presence'. [21] Whereas the infamous Marcel trips up on stone, at this valence in the work of the *manoeuvre* we may trip up on the matter of sound (ours and others), or find ourselves un-recognisably sounding, awaiting the station and—in-

voicing (invoking)—calling out to the commerce of art from within art, ears half-turned to a somevoice else which may also be our own speaking voice estranged, always half-present through the figure of voice in the *manoeuvre*'s essential dialectics.

## 6. Echolocation

The admixture of contingent conversive (between stations) and purposive (at stations) vocalic acts, and their necessary 'half'-ness or (e)strange poverties, shape the work the *manoeuvre* towards the sonorous practise of its full staging, which is to say at its momentary collection at each nominated station and concertina away from that station, as well as being a mode of the performance of critique, in this case, from its very plications, of the 'commerce' of Venice. The map of 'stations' of *Vedute Manoeuvre* are all indicated by a single orientational performance instruction, which acts as a ghost in the machine, or, unvoiced guidance amongst the other vocalic aspects of the manoeuvre.

Each card of the *Vedute Manoeuvre* not only asks for the image-side to be held up to the *veduta*[22] in order that we can check our 'station', and where we perform through an act of doubled looking a condensed mode of critical trans-historical comparison, but also the cards create a different layer of voiced meaning and vocalic performance once a station has been decided upon. We turn to the 'text' side of the card, but consider it as an exercise of voice. Each card possesses, under its numerical 'station', a quotation to be voiced aloud at that station. The waymarkers of each station are thus not only visual (the Canaletto image; the viewpoint in Venice proper) but also vocalic (our listening and vocalising practise with regard to the quotation). Yet due to the conversive, dehomogenising, conversational process that frames this moment, there is particular focus on the quotation-reading, an advanced, deeper, mode of listening created through the movement towards the station, to these 'impermanent vocalic markers'.[23]

Voice marks the spot. The performance and question of voice becomes purposive, even as the voice melts into the Venetian air as soon as it is sounded. Each quotation is performed by the voice of a body eminently conscious of its process within the *manoeuvre*, which takes the textual material and enunciates it according to its own performance decisions (to speak, shout, whisper, sing . . . ), all orientated towards the *vedute* of a given station as well as by the group who are making the manoeuvre, all working within the movements of the bodies of Venice that are not the manoeuvre. In the too-staged moments that were performed at the Venice Biennale of this manoeuvre, Brennan orchestrated a counterpoint of voices around the practise of quotation-interpretation: each practitioner-participant of the manoeuvre was invited to begin reading at different points, so that aspects of the quotation overlapped with each other—lines in sound as well as meaning (sense as well as sense) were drawn together to make this practise of voicing one which provided a different sense of what listening, what the act of giving voice (to perspective, to opinion) might be. Each voice acts on, through, with, and in part in ignorance of the next; the quotation become a kaleidoscope of respondent tonalities. We sound our own voice and hear it while listening to that of others and thus further estranged from the operations of our own voices and the voices of others through this staging we move between levels of vocalic self-estrangement, engaging in the extimate act of the voice. This movement of the *manoeuvre* allows us to think, to practise, resonance and response: what Dolar has identified as voice's 'break in presence'; how voice is 'a crack, a site, a locus, an opening which circumscribes the site of both subjectification and jouissance', is an excess, whose locus is resonance. Distinguishable from the other voicings—our voice at each station is neither one nor the next—voice in the *manoeuvre* simultaneously reverberates and caesures.

And we also try to *look* as well as *hear* deeply: the specificity of station-quotation derails vision—the hermeneutic mode seeks 'clues;' in the view, which is then derailed by the practice of concentrating on the voicing and meaning of the quotations. If we listen too hard, we turn away from the view; if we look too hard, we are distracted from the voice which speaks, or ventriloquises, the quotation. Each quotation provides a new, slantways, perspective on the visual matter at hand. The manoeuvre's voicings are much

more than conversive or pedagogical, they give rise to a consciousness of the voice as voice; of the speed at which words may be mistaken, voices misheard or eavesdropped upon, and how quickly consciousness of such an aspect of the performance, such a staged moment, becomes a self-consciousness, and how quickly the voice is lost in surrounding sound-world—its dimensions, its resonances, and effects never fully recordable (the voice object objects; an (im)materialist critique).

## 7. Vision-Trace-Occlusion

*Vedute Manoeuvre* begins in its material form and emplacement, moves to its conversational and object voicings, and subsequently—although these valences are introduced in very quick succession as the *manoeuvre* moves from intention to performance—on the act and occulting arts, of the visual. Whereas many of the performance documents of Brennan's *manoeuvres* provide either a consistent series of text-based route-guidance instructions, or are in themselves, formally, guide-books, here, the cards 'Station 1' and 'Station 2' give the performance instructions sparsely and thusly:

Instructions: Start. Outside Museo Correr, Piazza San Marco. Read Station 1 below. ('Station 1')

Instructions: Use the Canaletto reproductions as a means of orientating your position in Piazza San Marco. Tally the images with reality. Turn each over and read the accompanying quotations: ('Station 2')

The question of order is disrupted, and in so doing the *manoeuvre* allows each participant to question in turn the preconditions they have imposed on the idea of the start or end of this and of any participatory performance act. Thus, the 'art' and 'act' of manoeuvre elides with not only the anticipatory mode of preparatory movement towards, but also with basic principle of quotidian mobility. The work 'begins', like our strange response to the idea of 'Venice', multiple times. We are and are not actors within this work, as it, in turn, demands the movement between the directed and non-directed, as well as the consciousness of movement made theatrical or overly human.[24] The orientative mode of the work is continually, also, distracted and obscured by our surroundings and the guidance of the eye, to which the work points and within which it cannot but exist (a form of spoliate distraction, where each part dwelt upon is at once a part of a spatial continuum but represents also a chronic miscellany. 'Station 1' indicates that it is at once a beginning to the manoeuvre, has had a precursor manoeuvre (indeed is necessitated by a prior gathering and movement), and is in itself sort of precursor to the puzzle-work of orientation that each of the subsequent stations perform via the two-fold of the *vedute* (this '1' is a moment of judgement, a caesura which necessitates its own representation to mark the sequence it precipitates). And so, this station starts with a peculiar demand of our vision (or revision).

At this beginning which is simultaneously a non-beginning (does the command or the image, the curation or card, the emplacement in place or through voice, shape the start of the work?) the idea of the obscure prevails. Yet this is not as clichéd as it may seem thus stated. We start by recognising the *Museo Correr*, and therefore (in recognising the façade of the museum) know we may start (look back to the syntax and ordering of this preliminary command). We start on a threshold, a limens. But looking *at* the façade, we will later discover, we do not move *in*, but move *out*, in order to move *into* more fully the *manoeuvre*. But our first turn of the kaleidoscope is the façade itself, by which we orientate ourselves and in which orientation the spoliate mode of the work is first materially exposed. This is to write that St Marks is a battleground of medieval, early modern, and modern imperial de-historicising, re-aestheticising modes of spoliate display, in which the Correr (built in the 1830s) is indubitably a part of the highly politicised action of a Napoleonic-spoliate mode, whose presence as a relatively 'new' architectural aspect in itself frames and occludes what is pre- and post- the Napoleonic era in the piazza (as well as the objects and stony fragments that have been ex- and re-patriated into the space, objects of empire become objects of history, thence objects or waymarkers on the 'Grand' tour, then of the reproductive forces of the tourism industry),[25] and whose foundation, after the death of Teodoro Correr, was

the beginning of the Venetian state museum system. The first Station of the *manoeuvre* asks of the art-historical and political imaginary an act of Yatesean memory-work provoked by spoliate eye-rhymes, in order that the orientation and commentary at the station may be effected through a number of its potential dimensions.[26] To dwell here too long would be to be lost in a series of four-dimensional tracing-patterns, caught up only in dashes.

The act of the *manoeuvre* to provoke the eye to trace its surroundings in order to orientate the body within a given station, but also to orientate the station in an historiographical mode which may then precipitate a series of spoliate, self-obscuring visuocentric movements and/or commentaries (as above), is emphasised in the guiding image of this first of our fourteen stations—a photograph of a camera obscura apparently owned by Canaletto. Thus, the first 'Station' exposes a method behind the *vedute*. We enter into, or already are within, both a space of apocryphal ownership and a 'dark chamber', a world of potential misalignment and traced lines, or this form of projection is a comment on how we engage with the manoeuvre's precision, condensation, and deformations (evidence of Canaletto's use is in a drawing which is not a part of *Vedute Manoeuvre*—the 'View of Santa Maria Formosa toward the Right Side of the Square', in which the top of the bell tower is misaligned in the tracing).[27] Another method of the *manoeuvre*, exposed in this 'camera obscura' station emerges through the reproduction image of the camera obscura in relation to the movement between Stations 1 and 2: the image points towards the content of the museum, which we move away from.[28] Explanandum is occult, occluded, oblique. The visuocentric is, after all, projective; the projective becomes a method of punctum and condensation within the *manoeuvre*;[29] adds to the excessive object-life of each aspect of the work, to the timescales played out in the action. Indeed, the arcane future framings of the card-work itself is called to through the quotation on Station 1, which references the (now incomplete, apocryphally originary) Venetian Tarot.

## 8. Obscura-Gesture-Digression

The *camera obscura* as method for manoeuvre implies the visuocentric's relation to the trace, and the reproductive forms of 'Venice', or of the idea of the *veduta* (two forms of *reducere*), multiplied within the many-bodied participatory method of the *manoeuvre*. Whereas the *obscura* heralds the non-beginning of *Vedute Manoeuvre*, the second station is the first in the *vedute* sequence (see quotation above). The remaining 13 cards, or Stations, accede to the instruction given on Station 2, which means that the gestus—a combination of the visual and ocular—reigns as orientative. A counterpoint to this, the voice-work of the *manoeuvre* draws the performance-group together in the work. And, as with the voice-work of the manoeuvre, the visuocentric or ocular aspects move between the direct (*at* the Station) and indirect (*in between* the Stations), close (holding the card up to the eye) and distant (holding the card up to the *veduta*). *Vedute manoeuvre* asks its performers to find each Station through comparison with the Canaletto images. Orientation is effected through visual comparison *and* conversation (disquietudes that the performers of the manoeuvre have not quite gotten the right spot in which to stage each ventriloquial quotation lends itself to conversation—each aspect of the manoeuvre is contingent, moves into each other at the slightest possible dislocative point); orientation becomes a problem held in common within the manoeuvre; mis-orientation and curiosity, too, lend themselves to acts of recalibration and realisation.

As Brennan himself articulates of this practise within a different manoeuvre: ' . . . you can stop and look out, and you do, and you stop [ . . . ] and walk over to the edge and you see where you are, where all those little myopic turns have taken you.'[30] This is a performance practice which mobilises the liminal, the myopic, and pit them against expectation. In Venice, during *Vedute Manoeuvre*, we orientate ourselves through visual comparison-work; the turning of the cards, the movement of the head to look, to look again. But this comparison is only effective orientatively when we pay attention, almost solely, to the stonework. All else in the Canaletto reproductions, and in the surrounding 'Venice' of the time of the manoeuvre is mobile, mutable; the figures, the decorations, the

famous skies; the dog and the monkey in the piazza of Station 2 are and are not there (this, incidentally, is the station that most mirrors Vernet's spoliation documentation. Tellingly, the text of this card is taken from Walter Benjamin's 'Work of Art in the Age of Mechanical Reproduction'. On hearing this, the phonescreens which move humans through the square become further visible. The visuo-vocal of the *manoeuvre* demonstrates vividly how the form of the *manoeuvre* resists the trivial relationship between the visual and the real; the vocalic and authentic, sitting untidily alongside a *mise-en-abyme* whose mutable c.21st human elements aggressively display the digital practice of recording the body in space (the selfie, the tiktok, the 360-degree-pan of the piazza-video, all accessions of vision to a digital lens as intermediary; all calling out a horrible replicatory false individuation: *this has never been so done before, by which I mean by me, myself*). The stones, spoliate, demonstrate an art of reduction, yet from these arise the possibilities of cross-temporal comparison, the untimely art of the redux.

## 9. Para-Text-Ures

Each quotation disrupts the previous quotation and that which will occur next. The quotations, as Station 1 instructs us, are a sort of script or score for voicing at each Station-point, yet there are no performance instructions for the quotations except that they be read. *Vedute Manoeuvre*, considered as text-object beyond the merely instructive provokes its own voicing: any logic of order is to be provided through the performance act itself, and the *verbivocovisual* it brings together. As the *manoeuvre*'s command and process with regard to visual orientation asks us to seek a form of orientation, so to then may we apply this to the quotation texts. These texts provide new perspectives, issue from divers' sources and times. Each quotation draws attention to elements of the manoeuvre outside of the purely textual, for instance, Station 1's quotation contains a description of the 'thick paper-like material' of the Venetian Tarot, which calls attention to the card-stock and smooth texture of *Vedute Manoeuvre*'s cards as well as casting an arcane shadow over their contents, Station 2's quotation from Walter Benjamin draw attention to recorded forms ('a photograph or a phonograph record') and their repetitions which calls attention to the live art of the *manoeuvre* against the tourist-acts of the 'real' life of the square, Station 3's quotation from Thomas Mann speaks to Venice's spoliate 'unreality', drawing out the odd layering effects of the manoeuvre's process, Station 4 adds to these spoliate estrangement-effects through informing us that Canaletto's commercial fame existed for the most part outside of Venice, as do, even now, the majority of his works, and so on.

Disruptive in their eclectic gathering of any singular generic condition, the quotations of *Vedute Manoeuvre* can be mobilised to allow for the *manoeuvre* to operate on a basic, pedagogical level.[31] But *Vedute Manoeuvre*, responsive to multiple media and medialities, moves in its complexity away from this. And indeed, its non-beginning in the contingent, arcane, and occlusive indeed imply that an aspect of its performance politics is to disrupt the linear didactic mode of the pedagogical or touristic show and tell. Further disruptive are the additional textual elements of each card, and the performance questions they raise. Which is to write that the 'body' text of the *manoeuvre* (the two instructions and 14 quotations) is not naked, but perhaps indeed too fully clothed.[32] There is of course the text's 'other' side—the images—but there is also further text—the 'other' on the 'same' side (of the card). These others comprise, on each card, the Station number, the proper reference for each quotation, the proper reference for each image, and segments from an essay 'Venice Ventriloquised', that runs along beneath the quotation on each card. Lines are drawn between quotation and essay, essay and picture reference (the latter of which gestures to the visual other side of the card, and thus prompts the act of re-turning). We must decide, and there is no indicator, how much of this text is a part of the vocalic aspect of the *manoeuvre*; how much of this text is relevant to our Stationary understanding, how much educates, provokes, promotes conversation; how much of this text is redundant. We are forced into further acts of critical engagement, of condensation-work, which brings us once more to confront our position within the *manoeuvre* and how we relate to the

knowledge and power of that position, as well as its relations to what it is not, which is simultaneously the other performance-generators within the manoeuvre, the material dimensions of the manoeuvre, and all external elements. We constantly renegotiate the thresholds of the work; are brought, through textual excesses, to realise again what is (still) there and not there in erratic waste-spaces of the piazza and its moveable forms.[33]

## 10. Text-Imacy

The text of *Vedute Manoeuvre* is and is not at once performance instruction and performance documentation. It calls to the concept of performance documentation, and designs a possibility of performance. Such is the nature of the artefact, *Vedute Manoeuvre*, before, within, and after the moment of its performance, that the text at once sets and is excessive to the possibility of precedent. Were we not in Venice, we could, possessing the cards, nevertheless read, 'imagine', and thus experience the manoeuvre, almost novelistically, and, nevertheless, with something the sort of frisson of complexities and decision-making, that the experience of the *manoeuvre* in situ, as live art, may give (the materials allow for this, as does the way in which the text-instructions act on the thinking brain); if we have experienced the manoeuvre already, the cards, taken out of *situ*, nevertheless also comprise the art-work, triggering different forms of memory-work and reactivation.[34] The art of documentation within the *manoeuvre*, at once post hoc and pre-reconstruction, is, in a sense, the art that is and which allows for the 'redux' itself. No digital recording art is equipped for the multi-dimensional task of orientation by eye and voice through surrounding, image, text (vocalised) and text (silently read), singly and in a group, that the *manoeuvre* demands, in which each aspect in some way occludes the other yet draws it closer. Even the text-objects on one side of the card provoke a similar move around and between in themselves. Like a sort of Mobius strip, the *manoeuvre*'s documentation-loop works resists its being recorded; each aspect turns out to the next and back in upon itself in a constant looping movement of ex-centric self-distancing in the multiple via of voice and vision the *manoeuvre* charts—it is, to use Lacan's spatialising term, demonstrative of an art of extimacy.

## 11. Para-Taxis

On the text-side of the fifth card, or in our reading-hearing, the Station's dislocative force takes a sonorous poetic modality, which mirrors the counterpoints of the visible-invisible, spoken-unspoken, material-vocalic, which form the intemperate moves of the *manoeuvre*'s extimacies (printed here is Wordsworth's sonnet 'On The Extinction of the Venetian Republic'); for the lyric gesture (and this is a paradigm of lyric: a Romantic Sonnet) is one which is simultaneously self-held and self-annihilating, an outward–inwardfoldedness. The sonnet itself follows the spoliate logic of the manoeuvre inasmuch as not only is its form an elegiac paratactic-redux in the extreme (the octave of the sonnet condenses through retrojective parataxis a the normative history of the Venetian republic), and is composed of internal disconnects (it is, for instance, an 'Italian' sonnet in rhyme, but holds via its grammatical-punctuative aspects the ghost of the 'English' sonnet's capitulative, heroic couplet; thus both aspects clash, partly negating each other in a formal-immaterial political staging), but it also stages another aspect of the *manoeuvre*'s essential modalities: the date and conjunctural reason for the sonnet's composition are at best debated, and at worst apocryphal.[35] As with the camera obscura, the author-punctum disappears, replaced only by the option of looking in a particular way.

> Men are we, and must grieve when even the Shade
>
> Of that which once was great is passed away. ('Station 5')

As the sonnet concludes in this quasi-couplet (quasi because it exists in rhetoric but not rhyme; is an alien rhetoric to the form the poem seems to ascribe to, thus denatures the forms' steady progression), the poet layers shadow on shadow; Venice, spoliate, emerges precisely between the lines. It can no longer be what was 'Once', which is what the work illuminates, rather, what 'it' 'is' exists in the memory of the memory (a typically

Wordsworthian technique). It is more Lethean than Alethean (the echoes through rhyme continue even as the sonnet ends: *fade, paid, Shade*, and *decay, day, away*). The paratactic shows us what is not there, which is what is essential to what *is*. Between the lines, and *as* the lines, the sonnet, placed at this moment of the *manoeuvre* displays its unconscious. What sort of slip-up, or trip, allows us to see the desire in the death-wish for (or of) Venice, as much as the wish for life? We put together the 'half of life' of this city—its stony presence, the relation of whose stones, even, are debatable as to provenance—with the macrohistories of the city's variously interpretable rises and falls and their undocumented spaces between. The sonnet's catalogue is elegiac, but its negations, shown in the immaterial and occluded, are what persists. They are also what is most slippery—airy rather than solid—what, like the object voice, display the complication of being otherwise; being that is not wholly to do with the question of life. Spoliate.

In the excesses of its half-of-life, Venice absorbs its detractors, forcing such recourse to parataxis as self-undermining tool of proof, even in its most vociferous detractors.[36] The paratactic move is more than a cataloguing, it is an act, or art, of proving both the limitations and the limitless (thus an art of the *limens*, the liminal, the threshold) through rhetorical gesture; a rhetorical gesture which, under the auspices of this analysis of the *manoeuvre* is simultaneously mimetic of and incorporated within the art of the *manoeuvre* itself. Parataxis points, or trips; provides a caesuring movement or punctum. The 'half of life', or stone on water on sky on stone of Venice, its rich poverty, is akin to the encapsulating move of the *manoeuvre*'s internal-external (or, extimate) self-recognition. Both elements call up the need to commit to the impossible catalogue; to be distracted significantly by the peripheral; to *list* (where to list is to wish for as much as to catalogue, and here the paratactic slips into the parapractic with ease), and to realise that to listen deeply to a single, or pair of, resonances (the intersection of two spoliated elements) is a form of poverty of experience. Yet, as we move through the *manoeuvre* we move *between* and nevertheless form this betweenness continually through the movements the *manoeuvre* demands of us (each side of the card; each aspect of text; each moment in vocalic singularity, conversation, counterpoint; each exposure, focus, comparison, disturbance . . . ). The manoeuvre as dash, or dashing; respondent to Venice as dash or dashing; the two cognate and therefore mutually self-expository and self-annihilating. Parataxis *fills in the gaps*. As Freud famously states in *Moses and Monotheism*, it is a concretising method of sorts, an encounter with the too-spectral once-ness of the historic forms 'Incomplete and dim memories of the past, which we call tradition, are a great incentive to the artist, for he is free to fill in the gaps according to the behests of his imagination and to form after his own purpose the image of the time he has undertaken to produce.'[37] So, the art here lies in the parataxis of the *manoeuvre*, in the spoliate forms: one cannot 'fill in' the gaps between spolia, nor between the items on the list. Our richnesses and poverty, here, are one and the same economic form: the *taxis* for whom the *para* is the predicate slips up and perhaps shows us more than we should wish for—from the proper (taxis as arrangement, ordering, disjunctive side-by-sideness), to the improper (taxatio as rating, valuation, commerce), and the downright wrong (the taxi, after all, moves us from place to place, or, *metaphorical*, transports from one to another station), all of which we engage with as a matter of course in our engagement with the Venice *Vedute Manoeuvre* calls into question.

## 12. Para-Praxis

And so parataxis gives way, sometimes even through a mode of parapraxis, to para-praxis, another way of noticing the obscure, or tripping up, or the gesture towards and away, the commerce of the dash. We know parapraxis more commonly as the Freudian 'slip', but in the performance of Venice's stones, the Proustian 'trip' reigns as an afformative mode of Venetian culture-jamming. The Venetian performance mode at face value is of course wilfully associative (as began this essay), with a tendency to tip into nostalgia for what is 'lost' in the surrounds of each spoliate moment, as Ruskin, in *Stones of Venice*, accusing even Renaissance addition (heaven forfend should we mention the Napoleonic!)

as irreversibly veiling the Gothic mode—a sort of silting, occulting degeneration through re-surfacing.[38] But Ruskin, and the simplistic associative-parapractic, is wrong: it derides what is *spoiled*, mistaking this for the *spoliate*—attempts preservation (stasis at all costs!) rather than movement. This is not to write that the spoliate mode is accelerationist, nor that the manoeuvre is, but that the attention to the slip-ups, mistakes, miscastings or miscegenations, and what the gaps or dashes between these can do or expose and how we expose ourselves by trying to fill in (unconsciously) the gaps, is the mode of the *manoeuvre*'s historical-immaterialist critique.

So too further moves the *manoeuvre*—through Venice, in voice and vision, and through Ruskin, to his translator, Proust. Ruskin himself is veiled (doubly so) in Marcel Proust's infamous Venetian parapraxis (and veiled critique of the Ruskinian attitude to the relation of Venice with its stones), which also layers the ghost on the ghost of Venice in its description of the fine, neurotic, art of forgetting (the memory of the memory in the memory—the 'nocturnal piazza' (Station 14 quotes from the *Guermantes Way*) which, impossible to find again, is nevertheless 'it'—through the modus obscura 'Venice' is made crystalline). And this is an art which has distinctly *manoeuverish* formation: Marcel, avoiding a car (parataxis!) trips backwards (the pratfalling 'Angel of History') on the uneven paving of the Guermantes mansion courtyard (piazza?). And, addicted, attempts to do so again, and again (*redux* and its failed movements). The *reducere* that a successful reprisal elicits goes by the name of *Venice*:

> every time that I merely repeated this movement I achieved nothing; but if I succeeded [ . . . ] in recapturing what I had felt when I first placed my feed on the ground in this way again the dazzling and indistinct vision fluttered near me [ . . . ] and almost at once I recognised the vision: it was Venice, of which my efforts to describe it and the supposed snapshots taken by my memory had never told me anything, but which the sensation which I had once experienced as I stood upon the two uneven stones in the baptistery of St Marks had, recurring a moment ago, restored . . . [39]

The key to both the critique of Ruskin's singular desire for a renegade Gothic, and to the Proustian *manoeuvre* here is the fact that the stones are *necessarily* uneven, *necessarily* are two. The space between which makes of the stones a non-uniformity allows for the trip-up, the eruption into consciousness, of the unconscious, or, a new 'drive' (not, then, the car, or taxi). This is the nonvisible which comprises spoliation's disjunctive object life, its ostranenious aesthetics. The parataxis points, or trips, becomes parapraxis which provides punctum (of which more in below); redux is a work of mobile concentration, not simple repetition, reproduction, or facile condensation, and in fact these latter 3 are all modes of assembly that the redux of the manouevre exposes . . . .

## 13. Crux Contra-Punctum

Re-mark, then: X marks the spot. Where, in the mobile rationale of the *manoeuvre*, a part of whose critique is to develop an art of noticing in its participant that demands attention paid to what nonvisible or invocalisable processes have been occluded, obscured, X may be voice or vision, or X may be a punctuative stand-in cognate with the nilling-dash (the strikethrough or crux of the *punctum delens*), or the 'spot', locoparticular, of each Station. The concept of the crux is one which is pointed, and which shapes the manoeuvre in multiple ways.

In the first instance, the crux, or punctum (X marks a spot, or hole), shows to us the proto-camera's gaze. We return, then, to the start: the camera obscura, whose rear-projective technique for the promotion of accurate tracing is the opposite of that of its younger sister, the *camera lucida*. Yet it is with the *camera lucida* we begin, as, in the book of that name Roland Barthes infamously defined an indubitably twentieth-century idea of the punctum ('that accident which pricks me, (but also bruises me, is poignant to me)') which works against the *studium* (the historico-culturally conditioned 'reading' of a photograph), and which, as we have essayed through the *Vedute manoeuvre* must seem in some ways

cognate with the Proustian trip—that accident that yields affective engagement. Yet, the ante-photographic nature of the manoeuvre sees it bring together and complicate this dualist dynamic. Predicated on the *obscura* rather than the *lucida* as mode of elucidation, the *manoeuvre* works with a different form of projection.

Since there is no counterpoint without alternate measure, what of the second instance of counterpoint in the formal shaping of the manoeuvre? The via *crucis* whose crux-based structure, as previously mentioned, is a structural principle of the *manoeuvre*. The question of Venice in terms of Venice 'worship' thus displays itself through the *manoeuvre*'s formal critiques, eliding with the way the action of the manoeuvre asks us to dwell on and at each station, and move in a sequence of 1–14, between them (where the 'betweens', the silent numbers or noisy conversational manoeuvres work in a counterpoint, beginning before the first Station, numbering 1–15). The numbering principle of the cards is of course symbolic; 'Venice' (via Canaletto) acts in counterpoint to the via *Crucis*. It is not a coincidence, thus, that the image from Station 1—the camera obscura—which we have read as one of the working immaterial-symbolic concepts of the *manoeuvre*'s work as well as its critical and aesthetic dimensions, sits in counterpoint with the first ikon of the *via*, which is not the walk but its predicate (*viz.* the condemnation of Christ, who only then takes up the cross, or crux (Station 2), and will die). Mobilising a Christian ikonographic contemplative practise as a form for the socio-political aesthetic critique the *manoeuvre* demands that its participant-performer attempts (essays, or charts), makes for a further de-temporalising, estranging, effect, as well as an allowance through this counterpoint for further valences of (comparative) critique. The history of modern British art tells us that the levelling of the idea of the 'ikon' has, not only with Brennan's practise, a resonance with a re-do-ing (*redux*) of the *reducere* of the ikon-contemplation process as a mode of publicly engaged mobile political art practise.[40] And in *Vedute Manoeuvre*, if we know the via *crucis* well, we are brought into comparison, and the hermetic reasoning that this can induce—we look for keys to reason in (or behind) the count. In this sense the *manoeuvre*'s dialectics are didactic. What, then, might we make of the numerical congruences, where, for instance, Station 6 sees Christ's face wiped (*via*) and/or an article about the interdiction of the hijab in Venice's museums (*vedute*); we hesitate between things, between the punctum that singular authorial attribution allows, begin to see veils everywhere (think of Ruskin on Venice; think of Proust on the veil of illusion, or of the mopping of his brow in his illness; think of the material necessary to build a camera obscura . . . ).

But the art of the manoeuvre, *obscura*, is precisely to demonstrate and destabilise this form of renegade hermeticism; its peculiar mode of im-materialist critique, as we have seen, has, like many of Brennan's *manoeuvres*, aims to expose the unattended to *obvious*, not the hidden, through provoking different *ways* of looking (different *via*, different *vedute*), all of which interact with the staging of the immaterial through the figure of voice. We are brought back to the spoliate practise of the performance (quite the opposite aesthetic mode to the hermetic), and see the moves between what Richard Brilliant has called *spolia in se* (material forms) and *spolia in re* (the redux—reuse—of the very concept of spolia in the non-physical or virtual (quotation and reproduction, for instance).[41] This very immaterial aspect is the crux of the matter of the *manoeuvre*'s spoliate process perhaps. We perform Venice's stones *in their presence* yet through immaterial forms. Number allows us to chart, compare, concentrate, condense, but also (because sequential) to notice the aspects which are essential but not numerated in (or between) the sequence, and to internally think through progression and return (*redux*). Number is also the count—the pace of the *manoeuvre*, and its accounting measures (temporal, economic, material). So it may be that number, then, is the crux of this crux—Proust's problem of the two stones and their space between with its material and immaterial effects. Extimate, the process is comprised of footprints, of voices, and is built out of the revisiting and reanimation of traces, of moments gone awry, in the eye, the ear, the full body, but leaves none of this; the ghostly economies of Venice are thus exposed. It is at once designed for its own replication (*redux*) but is thus

internally self-enfolding. The redux (again—) of the dash (again—) and its condensations (again—) is the manoeuvre of the manoeuvre. Here –.

## 14. Redux Comma Dash

The time of (performance) art is not simple, and neither is the time of art history. Both demand at their outset an art of condensation-reduction in order to order, or orientate (physically, historiographically, methodologically). The demand of the outset, or the decision, (we ask in the fifth part here where, then, does the beginning really begin?), which is in itself a break, or dash, is ill attended to in (histories of) art practice but which a reading under the auspices of the *redux* allows us to approach more clearly. Here—dashing—the always-already constructed praxis of (performance) art meets the always-already constructed praxis of art history in a work of mutual self-exposition. A pause on (yet another) threshold precipitates an orientative movement; rhythms *per cola et commata*; attempt; *manoeuvre*. Each, in line and against the other, demonstrates through the speculative mode of the essay, the speculative punctuative methods of the untimely redux, of the dislocative dash a possibility of writing art (history) otherwise.[42]

**Funding:** This research received no external funding.

**Institutional Review Board Statement:** Not applicable.

**Informed Consent Statement:** Not applicable.

**Data Availability Statement:** Not applicable.

**Conflicts of Interest:** The author declares no conflict of interest.

## Notes

1.   Bringing Warburg and Benjamin, great Modern 'readers' of different modes of formulation and temporalisation in art, the cultural imaginary, and capital forms, is intentional here. The *pathosformel* and the *gestus* are different, and have conceptual ground that is radically other to each other, but both aid the mode, herein performed in essay form, of the spoliate imaginary to which a 'stony' or lithic performance art may give rise. Venice is, of course, a very particular, peculiar, case study.

2.   Alexander Nagel and Christopher S. Wood, *Anachronic Renaissance* (New York: Zone, Nagel and Wood 2010): 'a virtual psychological process governed by the rhythms of recognition, connection, and interpretation' (356).

3.   Op. cit.

4.   Perhaps the most positivist apparent negation of Venice is Regis Debray's *Against Venice*. Tr. John Howe (London: Pushkin, Debray 2013).

5.   I take this from Robert Harbison's 'eccentric' reading of Venetian spaces, which nevertheless holds in its words strong echoes of Hölderlin's infamous 1805 lyric 'Hälfte des Lebens' (Harbison writes: 'The shabby stone fields of Venice have been deprived of something; to get this heightening and clarifying as on stage we have given up half of life, the earth and the trees'; the two stanzas of Hölderlin's 'Half of Life' elicit, in a lake-side setting, first trees, then earth, both lost through seasonal deprivation, and more apparently essential than both, water and walls). See *Eccentric Spaces* (Harbison [1977] 2000. Cambridge MA, The MIT Press), p. 59.

6.   Adrian Stokes, *Venice: An Aspect of Art* (London: Faber and Faber, Stokes 1945), p. 1.

7.   See (Buršić et al. 2007; Lazzarini 2012).

8.   (Barry 2016).

9.   I refer here to the infamous moment in Marcel Proust's *A La Recherche du Temps Perdu*, where, on tripping on a paving stone (with more than a little nod to Ruskin's Venetian writings, which Proust translated), a cascade of memory-work occurs, taking Marcel to Venice. I refer to this in my 'Venice Ventriloquized' (in *Vedute Manoeuvre* (York: Information As Material, Brennan and Yeung 2011)).

10.   I thank my companions for helping with the 2022 Venice Scheme in logistics and intellectual companionship, adding their own thoughts and resonances to my idea of a 'spoliate' performance practise, viewing what I could not view, walking or gondoliering where I could not, helping the refinement of the in-discipline of the process; 'the more duskily the better': AC, AF, FR, *grazie mille*.

11.   Tim Brennan, with essay 'Venice Ventriloquised' by Heather H. Yeung *Vedute Manoeuvre* (York: Information As Material, Brennan and Yeung 2011).

12.   The Knowledge' exhibition, cur. James Putnam. 54th Venice Biennale. Gervasuti Foundation 2011.

13    In 'Walk-On: 40 Years of Art Walking from Richard Long to Janet Cardiff' exhibition, cur. Cynthia Morrison-Bell and Alistair Robinson with Mike Collier and Janet Ross. Northern Gallery of Contemporary Art 2013.

14    With regard to the Joycean *verbivocovisual* and the Kristevan kaleidoscopic and in particular with reference to Brennan's use of the polyvalences of the text-image-essay-command based card-work, see Heather H. Yeung, catalog essay for *New + Original Works* (Arucad Gallery, Brennan and Yeung 2019) n.p.

15    The material work named *Vedute Manoeuvre*, published by Information As Material for the Venice Biennale exhibition-performance and later re-exhibited as a part of the *Walk On* exhibition, has no page numbers (as it consists of a series of cards held together in a cardboard wallet); when referring to or quoting from particular cards I shall hereforth refer in-text to the 'station' number set on the card. These run 1–14.

16    Stokes, op.cit.. I've written elsewhere of this effect under the auspices of an exploration of Venice and critical whiteness through the mimetic migration into Venice-related material of the colonial, shock, and framing effects of *pietra Istria* in Venice itself. See (Yeung forthcoming).

17    Brennan's practise method of the *manoeuvre* spans more subject matter and material form than *Vedute Manoeuvre*, but across all of these the artist is clear that the material form which accompanies the *manoeuvre* (in any of its performance versions—private or public, singular or in groups) is indicative of possibility rather than prescription. The practice of choice is important in the performance of these works in each of the dimensions the works mobilise (the starting point for a list of theses dimensions would be: the textual, the visual, the vocalic, the material).

18    (Myers 2010).

19    Janet Hand, *Monograph: Tim Brennan* (York: Information as Material, Hand 2022), p. 31.

20    Pauline Oliveros *Deep Listening: A Composer's Sound Practice* (Lincoln NE: Deep Listening Publications, Oliveros 2005) outlines the method, for the most part, although the introduction gives aspects of the conceptual-practical mode.

21    (Dolar 2006).

22    Performance images from the original (incomplete) biennale *Vedute Manoeuvre* can (at time of writing) be found at the online teaching resources for 'Walking As Artistic Practice' (Müller n.d.).https://teaching.ellenmueller.com/walking/2021/11/28/tim-brennan-vedute-manoeuvre-2011/ (accessed 10 August 2022). Later in this essay, we find a consideration of the impossibility of common digital documentation vis à vis the practise of the *manoeuvre*.

23    See (Yeung n.d.).

24    The echo of Stanislavski's directive-performative composition of the 'superobjective' of the work is here intentional. See (Stanislavski [1937] 2013).

25    See (Namer 2022).

26    It would be interesting to place, alongside the Canaletto prints which comprise the *Vedute Manoeuvre* stations 2–14, the Venetian scenes in Carle Vernet's *Tableux histooriques des campagnes d'Italie*, whose *vedute* demonstrate the Napoleonic spoliation of Italy in action. 'Entrée des Français à Venise', in particular, is an image absolutely cognate with an interstitial view of the *piazza San Marco* afforded during the orientative *manoeuvre* between Brennan's Station 2 and 3. On the politics and memory-work of the Napoleonic spoliate method, see (Karrels 2018).

27    Philip Stedman's systematic comparison of Canaletto's vedute with photographs (funded by the British Academy in 2021 and ongoing at the time of writing) provides even more accurate proofs of the ways in which Canaletto used the camera oscuro as part of the way he 'formed' Venice in both the *vedute* and the *capricci*.

28    The *camera obscura* apparently used by Canaletto (an ownership inscription reads 'A. CANAL') is held in the Museo Correr, inv. Cl. XXIX sn.30.

29    (Directed) vision as universal punctum—see (Davidson 2003).

30    Tim Brennan, *Coals to Sunderland* manoeuvre. A/V Festival 2022.

31    Juliet Sprake reads Brennan's *manoeuvres* precisely thus: as pedagogical innovation in the guide-book form through the act of quotation as 'critical guiding practice'. See (Sprake 2016).

32    I reference here Gérard Genette's 'Introduction to the Paratext', whose second sentence reads that the literary work as text 'rarely appears in its naked state, without the reinforcement and accompaniment of a certain number of productions, themselves verbal or not . . . '. Tr. Marie Maclean. *NLH* 22.2 (Genette 1991), p. 261.

33    This is yet another aspect of the Venetian 'redux' as *arte povera* or 'half of life', of which Harbison writes: 'Though there is less space to waste, unlike other town squares there are genuinely left-over bits of space, so occupied more unevenly and erratically. The pure calculation of paving is undone by accident of wells or buildings thrown up in the middle; even statues in Venice look movable.' Op. Cit. 59. The objective question of Venetian 'moveability' has already been indicated both through allusion to the *pietra istria* and the Napoleonic acts of plunder; its fictionalisation visible in Canaletto's not-quite-accurate *vedute*.

34    With regard to the link between documentation and the concept of imagined, critical, performance 'reactivation' I follow Paul Auslander (see Auslander 2018), but nevertheless with the proviso that this and many of Brennan's *manoeuvres* are performatively self-doubling: there is the 'live art' performance form, as well as the artists book form, which is integrated into the 'live' experience

but can also be otherwise performed. *Vedute Manoeuvre*, indeed, stages the 'reactivist' mode, by taking that precise moment from Benjamin in 'Station 2'.

35 The sonnet was for a long time considered one of his 'Sonnets to Liberty' written 6 years after the treaty of Campo Franco, but the date is debated (see for instance, Hill 1979); it may or may not have been written in conversation with a copy of English Translation of Vittorio Barzoni's *An Accurate Account of the Fall of the Republic of Venice, and of the circumstances attending that event: in which the French system of undermining and revolutionizing states is exposed; and the true character of Buonaparte faithfully portrayed* (1804–1805), on loan (again with debated dates) from Samuel Taylor Coleridge.

36 For instance, Debray's *Against Venice* is precisely this: a catalogue of what the writer *dislikes* in Venice, which nevertheless is a sort of positivism: it nevertheless catalogues Venice. The psychoanalytically inclined amongst us on reading such a vociferous catalogue of dislike—precisely not an obviation—might see in it a reaction formulation.

37 (Freud 1939).

38 The 'veiling' motif also—obviously—brings with it multiple critical issues with regard to Ruskin's (ab)uses of the feminine and the oriental. Here, I have found Anuradha Chatterjee's *John Ruskin and the Fabric of Architecture* (Routledge, Chatterjee 2019) a measured, elegant intervention on a subject that can (given the life of the writer) easily provoke a far more reactionary response.

39 (Proust [1927] 2000).

40 See the history of the naming of the Ikon Gallery, Birmingham, from a period long before the project was a static art museum (see Watkins and Stevenson 2004).

41 See for a great précis of Brilliant's coinages the introduction to *Reuse Value: Spolia and Appropriation in Art and Architecture*. Ed. Richard Brilliant and Dale Kinney (Routledge, Brilliant and Kinney [2011] 2016).

42 I thank my (anonymous) reviewers for providing an orientative clarity in their readings of the draft of this work which allows it to, retaining the mode of the speculative sentence, to nevertheless conclude without entirely eating its own tail.

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

Yeung, Heather H. Forthcoming. 'Adrian Stokes' Aesthetics of Whiteness. *Modernity's Self, Modernism's Other: Marking Whiteness*, Edited by Sonita Sarkar and Jennifer Nesbitt. *in press*.

Yeung, Heather H. n.d. Tim Brennan: Taking Coals to Sunderland. Available online: www.walk.uk.net/portfolio/taking-coals-to-sunderland (accessed on 1 August 2022).