# Peer review of "Performing Venice’s Stones: Vedute Manoeuvre Redux"

_arts_

Round 1
Reviewer 1 Report
This is a remarkable piece, and not academic in spirit, but more of a writerly performance. While there are a considerable number of academic references, there are far more intellectual references, games, rephrasings, poetic and conceptual gestures. To read such a text is play, or perhaps better an event, and in this sense the author seems to be attempting to generate a praxis, or work, that is equivalent to the piece he analyzes here, Tim Brennan's Vedute Manoeuvre. (Why on earth they have only written thirteen sections, when there are 14 stages to the original work?)
This is one of the richest texts on the art (and literary) history referenced by a work of contemporary art I have ever encountered. It seeks to replicate also the kaleidoscopic dimensions of the city itself and of Brennan's work and I feel that it is rather successful in this regard. While there are many jabs and feints at making meaning in this text, what it lacks is a conclusion (Section 14?) that gives us a sense of the distillate that the author has accrued from this performative article. While the writer does land the reader at the end, imagine them now peeled from the city, the re-performance of the piece, back in their office or study trying to figure out what such an engagement might offer, whether to their colleagues or students reflecting on the sense (the second one yes) that can be thus derived from such an exciting and wonderful romp! (An afterword?) What does it add up to? Is this a memoir in which the experience and the various references it unleashed is the point itself, a kind of onanist monument to the memory of lines of flight? Is it a tribute to the displacement of art and city that Brennan intends--in which case, who/what is the hero so honored? Or is it rather a proposal for writing art and history otherwise, for relaying a sense that becomes essential, not as meaning but as a living breathing form that talks back, refutes the interpretations, and unhinges the praxis of the history of art, which was always a construction anyway?
There are no technical considerations I would transmit but the end left me hanging and there will be no sequel. So perhaps build a bridge back to a place where the reader can stand and perceive, among these multiple ruins, some significance in the exercise. Not to close the door, but to open it further.
Author Response
Thank you for taking the time and care to read this piece, and for the engaged, warm response, which means a lot -- the work I have been doing on and around Venice is some of the most difficult writing I've so far undertaken, mostly for reasons you outline in your review, in fact, and so for this particular essay and its task (the response to the variant hyper-visible economies of Venice and to Brennan's *Vedute Manoeuvre*) the performance essay seemed the most apposite mode. I write with some relief that this seems to have come off.
Regarding the numbering of the piece - I think I have been rather scuppered by journal-style conventions here, as the abstract was originally the point-zero (hence 14th station) of the work. I see that this does not work! Piqued, too, by the comment you made on the work's potential onanistic feel, I have added a short 14th station which works, I hope, as invitation, summation, and re-affirmation of the work's work, a sense of it's being something of an articulation of a method of 'reading' (and thence of writing) art/history otherwise. Thank you for drawing this out from the work; it helped me clarify my own perspective on the piece.
Reviewer 2 Report
We have opportunity to ‘read’ Venice and to be reminded of some iconic pieces about this eternal enigma, the desire and theme of architects, urban planners, artists, merchants, travelers – the city of Venice. While reading this text, we can easily get lost in nostalgia and in the past. However, as much as it belongs to the past, Venice is also the future, what this text reminds us of. We also read in this paper about the manoeuvre through Venice. This reading of the city might be understood in several ways. The value of this text is in its ambiguous interpretation where performance art, such as Vedute Manoeuvre, offers new readings.
Author Response
Thank you for this review. I have read it with care, and re-read the original submitted text with reference to it. I've not made substantive changes in the light of this review, but have added a 14th station - a short coda, if you will - in order to tie up the numerical references the piece offers (this, in the light of another review report). Thank you again for the report, and for taking time to read the essay.